# A Literature Review of Selected Bacterial Diseases in Alpacas and Llamas—Epidemiology, Clinical Signs and Diagnostics

**DOI:** 10.3390/ani14010045

**Published:** 2023-12-21

**Authors:** Kacper Konieczny, Małgorzata Pomorska-Mól

**Affiliations:** 1Department of Internal Diseases and Diagnostics, Poznan University of Life Sciences, Wolynska 35, 60-637 Poznan, Poland; kacper.konieczny@up.poznan.pl; 2Department of Preclinical Sciences and Infectious Diseases, Poznan University of Life Sciences, Wolynska 35, 60-637 Poznan, Poland

**Keywords:** llama, alpaca, South American Camelids, zoonoses

## Abstract

**Simple Summary:**

Over the last decade or more, New World Camelids, particularly llamas and alpacas have become popular farm animals all over the world. Many owners consider these animals as pets, resulting in closer contact between them and humans. Therefore, in spite of their relatively high resistance to infectious diseases, it is important to improve knowledge of, among other things, bacterial diseases affecting New World Camelids. This review aims to bring together the latest knowledge of diseases caused by *Clostridium* spp., *Mycobacterium tuberculosis* complex, *Mycobacterium avium* subsp. *paratuberculosis*, *Streptococcus* spp., *Escherichia coli*, *Pasteurella multocida*, *Manheimia haemolytica* and *Corynebacterium pseudotuberculosis* in llamas and alpacas.

**Abstract:**

The breeding of South American Camelids (SACs), particularly llamas and alpacas, is becoming increasingly popular in regions that are not their natural habitat, including Europe. These animals are considered to be relatively disease resistant. However, due to their growing popularity, special attention should be given to infections in llamas and alpacas. Knowledge of bacterial infections is very important to veterinarians and breeders. Many of these diseases also have zoonotic potential, so these animals must be considered as sources of potential zoonotic infections. Due to the limited information on many diseases occurring in llamas and alpacas, veterinarians often rely on data collected in other animal species, focusing on cattle, sheep and horses. This work aims to summarise the knowledge of diseases caused by *Clostridium* spp., *Mycobacterium tuberculosis* complex, *Mycobacterium avium* subsp. *paratuberculosis*, *Streptococcus* spp., *Escherichia coli*, *Pasteurella multocida*, *Manheimia haemolytica* and *Corynebacterium pseudotuberculosis* in llamas and alpacas, with particular attention to epidemiology, clinical signs and diagnostics.

## 1. Introduction

The breeding and raising of South American Camelids (SACs) are becoming increasingly popular in Europe. Camelids are economically and socially important for millions of people worldwide as a source of protein, hypoallergenic fibre, transportation, and manure for natural soil fertilisation. In addition, they are bred as companion and therapy animals [1].

The Camelidae family is divided between two tribes, Lamini, New World Camelids (NWCs) or South American Camelids (SACs), and Camelini, Old World Camelids (OWCs). OWCs consist of two species: the one-humped dromedary camel (*Camelus dromedarius*) and two-humped Bactrian camel (*Camelus bactrianus*). These species are large, even-toed, ungulates native to Asia and Africa. Indigenous to South America, NWCs are divided into four species: guanaco (*Lama guanicoe*), llama (*L. glama*), alpaca (*L. pacos*) and vicuña (*L. vicugna*) [2]. The species that comprise the Lamini tribe have a dual origin. Guanaco and vicuña are wild animals, while llamas and alpacas are domestic animals created with the cooperation of people and nature more than 5000 years ago in the Andes [3]. The original distribution range of species belonging to NWCs is the high-altitude grasslands, the Altiplano, and the Patagonian arid steppes [4]. One of the most significant advantages of alpacas and llamas is that they graze the grass without damaging the pasture structure. This is due to their relatively low body weight and specific digit structure [4]. Camelids are considered to be very resilient animals. However, many diseases, including infectious diseases, cause a loss of productivity and are important in camelid breeding. Their incidence depends mainly on climate and management conditions [5,6]. Over the past decades, the population of NWCs in Europe has grown, especially for alpacas. These animals are mainly raised for their excellent quality fibre, but also as companion animals and as tourist attraction [4]. The market for meat products from NWCs is also developing slowly [7]. Due to their appearance, gentle temperament and predisposition for training, alpacas are used extensively for animal-assisted therapy. Therapy using these animals can be effectively applied to patients with neurosis, depression, anxiety disorders, mental diseases and children with attention deficit hyperactivity disorder (ADHD), autism, or cerebral palsy [8]. With the growing and breeding of NWCs in Europe and worldwide, knowledge of diseases occurring in these species is becoming extremely important for veterinarians, pet owners, and breeders. The zoonotic potential of some bacterial infections affecting both domestic and wild NWCs should also be borne in mind. Humans can acquire zoonotic infections from these animals via direct or indirect contact with the animals and their environment, or via the oral route [6,7].

The following international databases were used to create the review: PubMed, Web of Science and Google Scholar. Articles were searched using key terms and phrases in combination or separate using “AND”/OR” such as “bacterial diseases”, “infectious diseases”, “South American Camelids”, “New World Camelids”, “Clostridium”, “Clostridium perfringens”, “enterotoxaemia”, “tuberculosis”, “Mycobacterium” “diagnostics”, “prevalence” and “clinical signs”. The study was carried out from January to July 2023. Articles no older than 1990 were used, with the exception of a few older articles providing valuable data. Specialist textbooks on camelids were also used. The selection of bacterial diseases described in the review was based on the number of articles available on bacterial disease. The diseases for which the most references were found were selected and described. Additional criteria for the selection of the diseases were prevalence, occurrence in South America and/or North America and Europe, and potential zoonotic risk.

### Major Diseases of Bacterial Infections in Alpacas and Llamas

This review presents the latest knowledge of diseases caused by *Clostridium* spp., *Mycobacterium tuberculosis* complex, *Mycobacterium avium* subsp. *paratuberculosis*, *Streptococcus* spp., *Escherichia coli*, *Pasteurella multocida*, *Manheimia haemolytica* and *Corynebacterium pseudotuberculosis* in llamas and alpacas with case reports in the literature. Based on the available data, the epidemiology, clinical manifestations and diagnostics are described. The summary of the data regarding the selected bacterial infections of llamas and alpacas is shown in Table 1.

## 2. Disease Caused by *Clostridium* spp.

*Clostridium* spp. are Gram-positive, rod-shaped, anaerobic, spore-forming bacteria present ubiquitously in the environment worldwide [10,28]. Clostridia can cause a wide variety of different diseases, both in domestic and wild animals [1]. Botulism (*Clostridium botulinum*), tetanus (*C. tetani*), blackleg (*C. chauvei*), malignant oedema (*C. septicum*), black disease (*C. novyi* Type B), bacillary haemoglobinuria (*C. haemolyticum*), and type A, C and D enterotoxaemia (*C. perfringens* Type A, C, D) are clostridial diseases found in camelids [29]. The prevalence of clostridia in samples collected from NWCs has been investigated by Twomey et al. [15]. Clostridia were detected in approximately 2.5% of the samples (44/1765) and the most common cause of enterotoxaemia was *Clostridium perfringens*. The diagnosis of enterotoxaemia was based on the finding of sudden animal death and bacterial overgrowth sufficient to produce toxins [15]. Clostridial infections are economically important as they cause significant losses in livestock production [28].

Although *Clostridium perfringens* is a part of the normal microflora of the digestive system of healthy animals, it causes infections in animals and humans [1]. There are five types of *C. perfringens*—A, B, C, D and E. In general, *C. perfringens* can produce more than 15 different toxins. However, this classification is based on the lethal toxin produced by the type of bacteria (alpha, beta, epsilon and iota) [12,28]. *C. perfringens* types A, C and D are believed to cause watery, profuse diarrhoea, often leading to rapid death in young camelids [30].

Enterotoxaemia caused by *C. perfringens* type A is a common disease of newborn alpacas, particularly in Peru. The main virulence factor is alpha toxin [29]. This bacterium is an important pathogen in stressful circumstances and results in a high death rate mostly for crias less than 4 weeks old, most commonly between 8 and 35 days of age. Enterotoxigenic strains of *C. perfringens* type A are considered particularly lethal. They rapidly cause neurological signs and death in the infected animal [31]. Clinical signs vary from sudden death to colic, sometimes intestinal gas tympany, central nervous signs of convulsions and opisthotonos, shock and death. Post-mortem lesions may include pulmonary congestion and intestinal distention with watery fluid and gas [30], although diarrhea is not usually present. *C. perfringens* is known to cause intestinal necrosis and haemorrhage in mammals. It is considered that *C. perfringens* type A facilitated the systemic spread of *L. monocytogenes* [32].

*C. perfringens* type C enterotoxemia most commonly affects newborn camelids under two weeks old (10–40 days of age) [1]. One case of enterotoxaemia caused by *C. perfringens* type C was described by Kotorri et al. [12] in an Albanian zoo-park llama. Beta toxin is regarded as the main virulence factor of this pathogen [1]. The symptoms most commonly seen in llamas are loss of appetite, depression and disoriented, cramps and lying on one side of the body. An elevated body temperature and haemorrhagic diarrhoea are also observed [12]. The usual necropsy findings are of haemorrhagic enteritis of the small intestine (mainly ileum and jejunum), with blood-stained intestinal contents and necrosis in a dark red color. The intestines are distended with gas and are severely congested. Pulmonary interstitial oedema and hydropericardium with a large amount of pericardial fluid containing fibrin are often seen. In addition, swelling and congestion of the liver, kidney and spleen are observed. There may be cerebral oedema and neuronal degeneration in the brain [12,29].

On the other hand, *C. perfringens* type D causes enterotoxaemia type D, an over-eating disease or pulpy kidney [30]. Epsilon toxin is the most important virulence factor of this type of *C. perfringens*. The disease is often associated with sudden animal deaths and central nervous system (CNS) symptoms such as convulsions, circling, prostration with opisthotonos and paddling, and coma [29].

Uzal et al. [13] reported a case of sudden death in a 2-month-old guanaco from Argentina due to infection with *Clostridium septicum* and *Clostridium novyi*. A necropsy was performed, during which a severe gelatinous haemorrhagic oedema was seen in the subcutaneous tissue and underlying muscles of the right flank and lumbar region. Petechiae on the epicardial surface were also visible. No penetrating wound was found in this case, but bruises were noted, which may indicate a previous injury [13]. *Clostridium septicum* can also be one of the aetiological agents of abscesses. St-Jean et al. [33] isolated this bacterium from a painful umbilical cord abscess in a llama.

Rosadio et al. [34] studied 108 intestinal samples from newborn alpacas with enterotoxemia and showed frequent co-infection with *Clostridium perfringens* and *Eimeria* spp. In 33 cases, representing 30.55% of the study group, *Eimeria macusaniensis* was found in addition to enterotoxemia caused by *Clostridium perfringens*. The results of this study indicated that damage to the intestinal mucosa caused by coccidia supports the overgrowth of *Clostridium perfringens* with toxin production, leading to fatal enterotoxemia [34,35].

In the diagnostics of *Clostridium* spp. infections, real-time PCR or bacterial culture, should be used in addition to clinical symptoms [11]. Protection against clostridial infection is provided by adequate colostrum uptake and IgG absorption from the vaccinated female. If adequate transfer of passive immunity has not occurred, the administration of clostridial antitoxin should be considered [36]. There is no effective treatment for many clostridial diseases; the only management method is vaccination. However, it should be noted that, in Europe, there is a lack of vaccines registered for use in SACs, and off-label vaccines are administered for other animals, most commonly sheep and cattle [37]. An animal’s tetanus vaccination status must be determined before procedures such as neutering. In case of doubt, tetanus antiserum should be given in addition to vaccination [38].

In crias from dams vaccinated and with an additional booster in the last third of pregnancy, a safe regimen is to start vaccination at 10–12 weeks of age. If the dam did not receive the booster during pregnancy, the vaccination cycle in crias should be started at 2–4 weeks of age, and a second dose should be given after 4–6 weeks [38]. Each animal should also be vaccinated against *Cl. perfringens* type C and D. The vaccination cycle starts with two doses given 4–6 weeks apart, followed by a booster every 6 months. Bentancor et al. [9] showed that llamas produce antibodies against *Clostridium perfringens* type D epsilon toxin after two inoculations 21 days apart. In areas with *Fasciola hepatica*, infections that result in increased susceptibility to clostridial infection, polyvalent vaccines are recommended [38].

## 3. Disease Caused by *Mycobacterium tuberculosis* Complex

Tuberculosis (TB) is a chronic, contagious disease of many vertebrate species [16]. It is caused by bacteria from the *Mycobacterium tuberculosis* complex (MTC), considered not species-specific, which is a primary agent of TB in humans. At the same time, *Mycobacterium bovis* is the primary pathogen of cattle but also causes TB in many other animals and humans [39]. Camelids are also susceptible to infection with *M. microti* [39,40]. It is acknowledged that NWCs are not highly susceptible to *Mycobacterium tuberculosis* complex infections. Still, in recent years, an increasing number of disease cases have been reported in these animals [41]. This is particularly true for the growing population of NWCs in the United Kingdom (UK), where TB is recognised as an emerging disease [39]. One possible reason for this phenomenon is the increasing maintenance of NWCs in areas endemic to TB [16]. Twomey et al. [15] reviewed laboratory submissions from NWCs supplied to laboratories in England and Wales between 2000 and 2011. They analysed 6757 submissions, of which 5154 were samples for disease diagnosis and 1765 for monitoring, and 3.2% (57/1765) of samples showed the presence of *Mycobacterium bovis* or *Mycobacterium microti* [15]. Between 2003 and 2009, in samples collected from 32 NWCs herds, *M. bovis* was isolated, while *M. microti* was isolated from samples collected from 10 herds [14]. More cases of TB in camelids are recorded among animals managed under intensive conditions and/or in close contact with cattle than those living in their natural habitat [42].

The interspecies transmission of infection to NWCs needs to be better understood. The non-camelid infection reservoir for NWCs consists of cattle and badgers [16]. The transmission mode among camelids is also unknown. It is presumed to be mainly horizontal via aerosol [14]. Other routes may be discharging skin lesions, faeces from enteric lesions, urine and congenital transfer [16]. The first case of bovine TB in llamas was confirmed in 1999 in a herd of five animals in the UK. *M. bovis* isolates obtained from animals from this herd were shown to have the same spoligotype as isolates from cattle and badgers from the surrounding area. This confirmed earlier speculation that llamas were susceptible to *M. bovis* infection and inter-species transmission of infection to NWCs can occur [43]. Thus, TB should always be considered in the differential diagnosis in the presence of non-specific respiratory signs or progressive weight loss resulting in extreme cachexia of the NWCs, especially in areas endemic for TB in cattle and wildlife [44]. The most common symptoms noted in NWCs with TB are wasting, anorexia, respiratory distress, enlargement of superficial lymph nodes, recumbency, discharged skin lesions and death [14,16]. Symptoms are usually associated with severe lung pathology. However, even in cases of severe pathological changes in the lungs, respiratory symptoms may not occur. Sometimes death is observed without previous symptoms [14].

Ante-mortem diagnosis of TB is challenging and often ineffective [45]. One method is the single intradermal comparative tuberculin test (SICTT). It uses a subcutaneous injection of 0.1 mL bovine and avian protein purified derivate at the axillary region. The skinfold thickness should be measured before and 72 h after tuberculin injection. In llamas, an increase in fold thickness of more than 2 mm or the presence of visible swelling at the site of injection of bovine tuberculin and, in the case of avian tuberculin, a thickening of the measured skin fold is considered a positive result [14]. However, disease control programs in NWCs based on tuberculin testing alone are mostly ineffective due to the very low sensitivity in these animals (14–20%) [16,46]. False-negative results are expected in SICTT, as indicated by negative test results with positive post-mortem examination and bacterial culture. False-positive results are also likely [40]. Another diagnostic method that can be used is blood sampling for serology using the lateral-flow rapid test (RT) or multiantigen print immunoassay (MAPIA). The lack of light chain in camelid antibodies, which makes forming immune complexes difficult, limits the sensitivity of lateral-flow assays [42,47]. A study carried out by Lyashchenko et al. [48] on a limited number of animals showed that serological tests, such as MAPIA and RT, allowed the ante-mortem correct determination of infection status (*M. microti*) in 97 and 87% of the animals tested, respectively. The above indicates the potential of serological methods for diagnosing *M. microti* in NWCs, but the small number of animals used in this study should be noted [48]. Infantes-Lorenzo et al. [46] have demonstrated that the P22 ELISA potentially provides a sensitive and specific platform for improved tuberculosis surveillance in camelids. This test has a 100% specificity and an average sensitivity of 62.5%. Implementing serological testing with SICTT increases the detection of infected animals [49]. For recognition of the clinical presentation of the disease, tuberculin and serological testing are valuable tools for controlling the outbreak but are unreliable in identifying individual cases of TB [14].

A definitive diagnosis of TB can only be confirmed by post-mortem examination, observing typical gross lesions and performing a histopathological examination of the lesions and bacterial culture [16]. Nodules develop mainly in the lungs, thoracic lymph nodes, and other organs, such as the liver, spleen, kidney, trachea, and pericardium [16,43,45]. Lesions may also be present on the skin, intestines/mesentery, and other lymph nodes [14]. The lesions that develop in the organs are histologically characterised by granuloma formation, with infiltration by epithelioid macrophages, lymphocytes, plasma cells and neutrophils, central caseous necrosis and variable calcification and fibroblastic reaction [16]. Langhans’ giant cells, typically found in tubercles, may be absent [14]. On post-mortem examination, an excessive amount of fluid in the abdominal cavity and an excessive amount of fluid together with fibrin in the pleural cavity can be observed [43]. Grape-like lesions may also occur on the parietal pleura, lungs and pericardial sac [43,45]. Ziehl–Neelsen staining of smears from lung and lymph node lesions reveals numerous acid-fast bacteria, suggesting the presence of mycobacteria. A smear can also be made from an exudate taken from skin lesions. However, relatively often, only a small number of strained bacteria are present, or a complete absence of bacteria [16,39,43,45].

Mycobacteria are slow-growing microorganisms requiring specific media, such as Lowenstein–Jensen or Ogawa [16]. A faster identification method is polymerase chain reaction (PCR) assays, for which fresh tissues are not necessary but can be formalin fixed. However, PCR has its disadvantages. These include low sensitivity, possibly associated with low numbers and uneven distribution of the bacteria in tissues, and no possibility of further discrimination of the isolate by molecular typing, which can be carried out using cultured isolates [16].

Human infection most often occurs through consuming unpasteurised cow’s milk and, less commonly, inhaling infected droplets [16]. In developed countries, *M. bovis*-associated human TB cases are relatively rare due to eradication programmes and pasteurisation of milk [50]. However, *M. bovis* still poses a zoonotic risk to people working with and handling infected animals [16]. Twomey et al. [39] described the first case of cutaneous TB in a veterinary surgeon. The infection probably occurred during the clinical examination and treatment of a 3-month-old alpaca cria. Analysis of samples taken from skin lesions of veterinarians confirmed the presence of *M. bovis* [39].

## 4. Disease Caused by *Mycobacterium avium* subsp. *paratuberculosis*

Johne’s disease (JD) or paratuberculosis is a bacterial disease caused by *Mycobacterium avium* subspecies *paratuberculosis* (MAP). The primary hosts are domestic ruminants, such as cattle, sheep, goats, alpacas, and llamas; however, MAP is a multi-host pathogen, and infection has also been detected in several wild ruminants [51,52]. MAP is highly resistant to environmental conditions, surviving on pasture for over six months. The most important transmission route is the faecal–oral route [51]. Prevention and control of MAP infections are complicated due to the long subclinical phase, the presence of few animals in the herd with clinical signs, the difficulty of diagnosing the early stage of infection and the high survival rate of the bacteria in the environment [53]. Due to the large number of bacteria excreted by cattle in their faeces (10^8^ bacteria per gram of faeces) and the incidence of infection in wild animals living near farming areas, cattle and other ruminants’ faeces are considered to be the primary source of infection in different animal species [18].

MAP infection in NWCs can cause granulomatous lesions in the small intestines, leading to protein-losing enteropathy with severe weight loss [17]. As with small ruminants, NWCs may show asymptomatic infections without diarrhoea and weight loss, making it difficult to detect in the herd [17,52]. In addition, in alpaca with paratuberculosis, non-specific symptoms include dullness, poor coat, pale and dry mucous membranes, dehydration and severe weight loss, resulting in life-threatening cachexia, although a preserved appetite and lack of diarrhoea have been observed [54]. Weight loss in alpacas and llamas can be masked by their bushy fleece, making it even more challenging to observe symptoms [17]. The presence of disease symptoms is often related to the shedding of bacteria with the faeces. The number of pathogens in the faeces indicates the progression and severity of the disease [51]. However, infected animals may shed MAP in their feces months or years before clinical signs of the disease appear. Therefore, asymptomatic shedders spread the pathogen [17]. The main necropsy findings occurring in NWCs with MAP infection are enlarged mesenteric lymph nodes, which on histopathology show extensive granulomatous lymphadenitis with areas of necrosis and mineralisation [55]. Thickened and corrugated mucosal surface of the distal small intestine and colon may occur [56]. In some cases, severe granulomatous enteritis and hepatitis were reported [55]. On histopathological examination, infiltration of the lamina propria by epithelial macrophages containing numerous acid-fast-staining organisms and plasma cells and lymphocytes can be observed [56].

The most critical requirement for effectively controlling paratuberculosis in the NWC population is early detection of MAP carriers and shedders, which are a constant source of infection for other animals [54]. In general, direct methods, such as PCR and bacterial culture (faecal samples), and indirect methods, such as ELISA (milk or serum samples), AGID, and complement fixation test (CFT), are helpful in the diagnosis of MAP infections [57]. Being the “gold standard” for MAP identification, bacterial cultures from faeces or tissues are not effective from a practical point of view due to the long growth and identification time (about 8–12 weeks) [53]. Samples can be cultured on Herrold’s egg yolk medium [58]. Serological tests, such as ELISA, can be worthy in laboratory diagnostics of MAP infection. However, the sensitivity of ELISAs varies from 50 to 70%, which is mainly because MAP is an intracellular pathogen and does not induce a strong humoral response from the body [17]. Miller et al. [53] showed a specificity of the different ELISA variations ranging from 48% to 92%. The G-protein conjugate ELISA may be useful for multi-species applications. The accuracy of ELISAs can be improved by using age- and sex-specific cut-off values. However, any positive serological result should be confirmed by culture. Therefore, ELISA is useful for evaluating the entire herd rather than confirming the disease in individual animals [53]. Real-time polymerase chain reaction (RT-PCR) is an accurate and rapid method for detecting MAP faecal shedding in NWCs. The sensitivity of this method ranges from 60 to 97% [17]. An additional diagnostic difficulty is the possibility of mis-differentiating MAP from other *Mycobacteria* species, for example, *Mycobacterium avium* subsp. *avium*. Lucas et al. [59] described a case of isolation of *M. avium* subsp. *avium* from an alpaca with symptoms resembling paratuberculosis. Infections with both bacteria can cause very similar clinical signs and histological changes [59].

Sporadic cases of JD in NWCs outside their native regions of origin have been reported; however, it was only in 2016 that the disease was demonstrated in alpacas in Chile [29]. Salgado et al. [18] conducted the first study to confirm the presence of MAP in a population of alpacas in their original habitat, the Chilean Altiplano. PCR detected fifteen (15/85) MAP-positive animals. It is presumed that the presence of MAP in the local population is due to environmental contamination and direct contact between other small ruminants and alpacas, which sometimes share pastures. Salgado et al. [58] conducted an infection prevalence study on 501 wild guanacos from Tierra del Fuego Island, Chile, and found that 21 of 501 individuals (4.2%) were MAP-positive. This was the first study demonstrating the presence of MAP among a wildlife population in Chile. Corti et al. [51] showed that sheep populations could be a significant source of infection for guanaco and other susceptible animals living on the Patagonian steppes. MAP circulates in both populations with self-sustaining transmission. MAP prevalence ranges from 5.0% to 92.5% depending on host species (sheep/guanaco), season and site type (shared/non-shared site). It has been shown that bacterial shedding in sheep is significantly higher than in guanaco [51].

Stanitznig et al. [52] analysed the prevalence of MAP in Austria. Of 399 faecal samples collected from NWCs, all were negative for MAP excretion in fixed faecal cultures, 3.8% (15/399) were positive by quantitative polymerase chain reaction (qPCR), 1.4% (6/443) of serum samples were positive by ELISA, and 0.2% (1/443) were doubtful. A study by Fecteau et al. [17], using faecal samples from four veterinary university hospitals in the United States (US), found a MAP prevalence of about of 6% in alpacas. Münster et al. [54] described a MAP infection case in an alpaca in a zoo in North Westphalia. The fact that the infection has been confirmed in Europe indicates the potential circulation of the pathogen among a population of NWCs in Europe, highlighting the need for routine monitoring for MAP infection in these animals. Controlling the incidence of Johne’s disease should be based on adequate colostrum intake of cria that come from negative dams and keeping the cria away from the female. Maintaining a clean rearing environment is also crucial. Preventive vaccination can also be performed [38].

## 5. Disease Caused by *Streptococcus* spp.

*Streptococcus equi* spp. *zooepidemicus* is a Gram-positive, beta-haemolytic Lancefield group C *Streptococcus* that causes acute, subacute and chronic infections in camelids [20,60]. It is considered a commensal organism of the digestive tract in NWCs inhabiting South America [21,61]. It causes two diseases, local and superficial, associated with wound infection or systemic, secondary to ingestion or possibly inhalation of the organism. The microorganism was isolated from blood, milk, uterine discharges, abscesses, wounds and pleural and peritoneal fluids in NWCs [19]. In North America, this bacterium is not considered a commensal organism [20]. *Streptococcus equi* spp. *zooepidemicus* is the aetiological agent of ‘Alpaca fever’ [21]. The bacterium was also isolated from cases of pneumonia in NWCs. *S. equi* spp. *zooepidemicus* was also reported in cases of mastitis in llamas with anorexia and pyrexia three days after delivery [21] and septic orchitis in alpacas [19]. Septic peritonitis is a significant cause of morbidity and mortality in NWCs [20]. In one study, it was responsible for 3% of deaths in these animals [61]. In Peru, the morbidity rate for Alpaca fever is estimated at 5–10% [20]. The disease can affect a wide age range of animals and has a high mortality rate [21]. In some South American countries, the mortality rate is 50–100% [20]. Stress factors, i.e., transport, shearing or processing, viral infections, high temperatures and malnutrition, can develop a systemic infection in subclinical carriers, or the infection is transmitted by aerogenic route from other camelids or other animal species [20,61]. Transmission occurs through contact with infectious material from carriers [21]. It is also thought that transmission of infection from horses to NWCs is possible. This bacterium is often isolated as a commensal organism from the respiratory tract of clinically healthy horses. Disease cases indicate this type of inter-species transmission in camelids kept together with horses [61]. Experimental endotracheal inoculation of alpacas with *S. equi* spp. *zooepidemicus* causes systemic disease and lesions typical for Alpaca fever, therefore aerogenic transmission is suggested [62]. The mechanical vector can be clothes and equipment used by people handling NWCs and other animals simultaneously, particularly horses [20]. Causes of septic peritonitis due to *S. equi* spp. *zooepidemicus* can be bacterial leakage from the injured gastrointestinal tract, direct inoculation through a penetrating wound, or blood-borne spread of bacteria into the abdominal cavity [61]. Young animals are predisposed to a systemic manifestation of the syndrome; however, the disease can also occur in adult animals [21,60].

Alpaca fever can present in acute, subacute and chronic forms [19], and usually develops within 24 h of infection. Alpaca fever is characterised by serositis of the thoracic and abdominal cavities with a high mortality rate. The most common manifestation of bacteraemia is peritonitis and pleuritis [21]. Necropsy can only show diffuse purulent peritonitis with extensive fibrous adhesions throughout the abdominal cavity [61]. In NWCs, lesions are limited to the serosal surfaces, and parenchymal organ lesions are sporadic. The development of bacteraemia can also lead to severe, diffuse, purulent meningitis [21]. Clinical signs of infection are non-specific and include recumbency, anorexia, hypothermia, tachycardia and tachypnea, lack of first compartment (C1) contractility, decreased faecal production, lethargy, tenesmus, and diarrhoea [21,61]. Marked bilateral ventral abdominal distention may be observed [61]. Anorexia, depression and fever occur in acute and subacute forms. The infection becomes systemic after ingestion of the bacteria, and death usually occurs 4–8 days after the onset of symptoms [19]. In the case of pulmonary infections, severe incidental pulmonary rales and pleural frictions may be found bilaterally on lung auscultation and an increase in bronchovesicular cranioventrally on both sides. On palpation, the animal may show pain reactions [21,61]. With septicaemia, cardiac murmurs can also be detected on auscultation. These usually indicate aortic valve insufficiency due to vegetative changes. On neurological examination, in cases of meningitis, vertical nystagmus, left facial nerve palsy, and occasional seizure-like activity can be found [21]. On haematological examination, changes typical of an acute inflammatory response are observed, such as leukocytosis, neutrophilia with a regenerative left shift and toxic degeneration of neutrophils, monocytosis and elevated fibrinogen levels [61]. On serum biochemical examination, hyperglycaemia (somewhat related to the stress caused by the examination) and hypoproteinaemia characterised by mild hypoalbuminaemia and hyperglobulinaemia, hyponatremia, hypochloremia, azotemia, and hyperphosphatemia are noted [21,61]. CSF testing is a valuable diagnostic tool in neurological cases. Grossly turbid and pink fluid is found, as well as neutrophilic pleocytosis, determined by increased protein concentration and increased total nucleated cells with 95% degenerated neutrophils. Culture and bacterial isolation can also be performed using samples from the brain, blood, CSF, lung and other affected tissues [21]. In the case of peritonitis, abdominal paracentesis yielded a turbid yellow fluid with a high nucleated cell count (80% granulocytes). A culture can be performed from the collected fluid to identify the bacteria [61]. Other additional examinations that can be carried out are ultrasound and X-ray. On ultrasound, thickened, hyperechoic pleura, aerated lung, and later peritoneal fluid with fibrous shreds are found in cases of pulmonary infection [21]. In cases of peritonitis, it is possible to observe a large volume of free peritoneal fluid with mixed echogenicity [61]. X-ray shows a caudoventral alveolar pattern typical of bronchopneumonia and pleuropneumonia [21,61].

In necropsy, possible findings include multiple strands of fibrin attached to the parietal and visceral pleura, sero-fibrinous exudate free in the thoracic cavity, congested edematous and collapsed lungs, endocardial and epicardial ecchymoses, petechial and ecchymotic haemorrhages of the congested abdominal serosa, and few intra-abdominal fibrin strands [20]. Histopathologically, fibrinocellular polyserositis can be observed affecting the peritoneum, pericardial sac, and pleura, and it is characterised by multifocal deposits of fibrin admixed with macrophages, lymphocytes, plasma cells and neutrophils. The lung parenchyma may be diffusely congested and oedematous with areas of atherectasia and haemorrhage. Alveolar spaces may contain fibrosis, neutrophils, macrophages, lymphocytes, plasma cells and Gram-positive cocci [20]. Intravascular coccoid bacteria are found in the lungs, spleen, adrenal glands, heart, kidneys and small intestine. In meningitis, histopathologically, purulent, multifocal and intravascular coccoid bacteria are noted in the brain’s grey matter. They can be isolated from the meninges using bacteriological culture [21].

In a case of septic orchitis described by Aubry et al. [19] in an alpaca, acute unilateral scrotal swelling, recumbency, anorexia, hypothermia and enlargement of the right testis and surrounding scrotum were observed on clinical examination. The right testicle was hard, warm, painful and immovable in the scrotum. Haematological examination showed neutrophilia with an increased number of band neutrophils and lymphopenia. A microscopic evaluation of the impression smear revealed many neutrophils and numerous Gram-positive cocci. Orchitis is thought to have occurred due to a chronic form of Alpaca fever [19]. Histopathology showed severe, multifocal to coalescing, suppurative, unilateral orchitis and epididymis with intravascular Gram-positive cocci.

Another *Streptococcus* species recorded in NWCs is *Streptococcus agalactiae* [63]. It belongs to the Lancefield Group B Streptococcus and is a pathogen affecting various animal species and humans [64]. In camels, it is responsible for developing mastitis, serous lymphadenitis, infectious skin necrosis and suppurative infections [65]. *S. agalactiae* is considered part of the microbiota of the respiratory, genital and gastrointestinal systems of humans and animals, particularly ruminants. A critical step in the pathogenesis of *S. agalactiae* infection is adherence to host epithelial cells, resulting in biofilm formation [66].

In one case, the only clinical signs observed after a suspected wound infection in the mandibular area of a llama by *S. agalactiae* were chronic suppurative subcutaneous infection in the intermandibular area and sudden death [63]. A post-mortem examination revealed a 2–3 cm abscess with a fistula in the intermandibular area filled with yellowish contents and diffuse subcutaneous oedema in this region, on the neck and in the caudal part of the head. Enlarged and swollen pharyngeal lymph nodes and severe sero-fibrinous ascites were also noted. Other findings were diffuse liver necrosis, high grade from sero-fibrinogenous effusion in the thoracic cavity, interstitial pneumonia, and swollen pericardium, and heart base. Intense infestation with gastrointestinal parasites may be a predisposing factor for streptococcal septicaemia [63]. The simultaneous presence of an abscess caused by *S. agalactiae* and infestation with gastrointestinal nematodes was most likely the cause of death in this animal [67]. On histopathological examination, diffuse mild to moderate neutrophilic and lymphocytic interstitial infiltration with alveolar emphysema in the lung parenchyma and multifocal necrosis of hepatocytes with foci of neutrophilic and lymphatic infiltration and diffuse hydropic degeneration of hepatocytes were detected [63]. Diagnosis of *S. agalactiae* infection can be performed by commercial kit or MALDI-TOF MS [68].

Volokhov et al. [69] described a new species of *Streptococcus* spp. They were isolated from alpaca faeces. They proposed the name *Streptococcus vicugnae* sp. *nov*, demonstrating the most remarkable similarity to *Streptococcus equinus*, *Streptococcus lutetiensis* and *Streptococcus infantarius*. This species is indole-, oxidase- and catalase-negative, non-spore-forming, non-motile, with Gram-positive cocci in short and long chains and facultative anaerobes [69]. Twomey et al. [44] described the first fatal fibrinopurulent meningoencephalitis associated with *Streptococcus bovis* infection in a 10-day-old cria alpaca. *S. bovis* belongs to the Lancefield Group D *Streptococcus*. It is considered to be an opportunistic pathogen inhabiting mainly the gastrointestinal tract. In this case, no premonitory clinical signs were noted in cria before sudden death. Necropsy showed the presence of fibrous material and semi-clotted milk in the gastric compartments. Biochemical tests identified the isolate as *S. bovis* biotype I. Histopathological examination detected diffuse meningeal infiltration with predominantly neutrophils and macrophages, intermixed with fibrinous exudate, mainly in the deep cerebral sulci, congested meningeal blood vessels and the presence of Gram-positive cocci in the meningeal exudate and the lumen of some vessels [44]. *Streptococcus* spp. may also be one of the aetiological agents of umbilical abscesses. St-Jean et al. [33] isolated *Streptococcus equisimilis* from the contents of an umbilical abscess in a 3-month-old llama. Other streptococcal species isolated from alpacas were *Streptococcus pyogenes*, *Streptococcus faecalis*, *Streptococcus equinus* and *Streptococcus uberis* [63,70]. The effectiveness of treatment is questionable. Treatment and drainage of local lesions in chronic cases should include systemic antibiotics, as superficial infections can become systemic [19]. Therapy for *S. equi* spp. *zooepidemicus* infections should be performed with intravenous fluid, an antibiotic, e.g., ceftiofur, ampicillin (due to suspected developing septicaemia) and a non-steroidal anti-inflammatory drug, e.g., flunixin meglumine (analgesic effect and anti-endotoxic effect) [21]. Hewson et Cebra [61] proposed a different treatment regimen for cases of peritonitis due to *S. equi* spp. *zooepidemicus*. They recommend sodium penicillin (22,000 IU/kg) and sodium ceftiofur (2 mg/kg). In case of insufficient animal response to antibiotic therapy, peritoneal lavage can be performed. In addition, the animal should be given fluids such as Lactated Ringer’s solution IV, supplemented with potassium chloride and sodium chloride [61]. A case of mastitis due to *S. equi* spp. *zooepidemicus* was cured with intramammary cephapirin and systemic antibiotics such as ampicillin [70]. Based on the Minimum Inhibitory concentration (MIC), the sensitivity of *S. agalactiae* to ampicillin and ceftiofur administered IV or IM in NWCs was determined. Therefore, using these antibiotics or administering antibiotics, selected based on an antibiogram performed on isolates cultured from an infected animal, is recommended for infections with this bacterium [63]. In the case of orchitis, castration is recommended. An antibiotic and a nonsteroidal anti-inflammatory drug (NSAID, for example, procaine penicillin G and flunixin meglumine) are recommended after the procedure [19]. In the case of intra-abdominal abscesses, antibiotic therapy alone will not usually result in complete absorption of the abscess. In such cases, a marsupialisation procedure can be performed [71]. If infection is recorded in a herd, given the mode of transmission of the infection, preventive treatment of the whole herd can be applied. The decision to treat the herd metaphylactically must be made after analysing the risks and benefits and determining the bacteria’s susceptibility to antimicrobials. The off-label long-acting ceftiofur is recommended for the metaphilactics treatment of the whole flock. However, it is always necessary to determine the microbial susceptibility of isolates and select the appropriate antibiotic on this basis. Chlortetracycline can also be added to the supplementary feed [21].

## 6. Disease Caused by *Escherichia coli*

*Escherichia coli* is a gram-negative bacterium commonly isolated from the faeces of many animal species [72]. *E. coli* strains show varying pathogenicity, causing diseases ranging from diarrhoea to lethal septicaemia (sepsis; colisepticaemia; systemic inflammatory response syndrome—SIRS) [73]. Strains in humans are divided into enterotoxigenic (ETEC), enterohaemorrhagic (EHEC), enteropathogenic (EPEC), enteroaggregative (EAggEC), diffuse adherence (DAEC) and entero-invasive (EIEC). This classification is based on the presence of genes responsible for producing various proteins [73]. In NWCs, EPEC and EHEC pathotypes are most common, alone or combined with *Cryptosporidium* spp. invasion and/or clostridial infections [73,74]. Mixed rotavirus infections with *E. coli* have also been shown to occur [75]. Featherstone et al. [76] showed a prevalence of VTEC (verocytotoxigenic *E. coli*) in alpacas of approximately 2%. VTEC O157 is transmitted asymptomatically among animals but can cause many diseases, including systemic diseases, in humans [76]. In another study, the prevalence of VTEC strains in alpacas and llamas between 1997 and 2007 was 4% and 9%, respectively [77]. STEC (Shiga toxin-producing *E. coli*) belonging to serotype O26:H11 was isolated from a guanaco with severe watery diarrhoea in Argentina [74].

A significant contribution to the development of neonatal diarrhoea is *E. coli* [72]. Neonatal diarrhoea is estimated to affect 23% of alpaca and llama crias. It occurs most often associated with septicemia and usually affects newborns at 3–7 days of age [74]. In a study by Dolente et al. [22], the prevalence of colisepticaemia was estimated at 17% of crias, and the mortality rate was 50%. Naturally, rotavirus occurs with *E. coli* biotypes and F41 fimbrial antigens in 18.8% of diarrhoea cases in crias [75]. In a study conducted in Peru, *E. coli* was isolated in 80% of diarrhoea-related deaths in young alpacas [78]. In other studies, in Peru, *E. coli* was isolated in 34% of diarrhoeal cases, with 60% of cases being co-infections (up to four different pathogens) [5]. *E. coli* can also cause meningoencephalitis and brain abscessation in NWCs [23].

The colonisation of target cells is made possible by the presence of fimbriae and other adhesins. The predominant fimbrial gene in EHEC and EPEC strains isolated from diarrhoeal cases in newborn alpacas is the F17 fimbriae gene [73], and 26% of *E. coli* isolates from neonatal NWCs with intestinal infection had the F41 antigen [75]. The intimin encoded by the *eae* gene enables adhesion to epithelial cells. *E. coli* strains carried Stx1 and *eae* genes, showed localised adherence to HEp-2 cells and produced entero-haemolysin [73]. Attaching and effacing *E. coli* are strains that attach to enterocytes and form attaching-effacing lesions (A/E lesions). These include EHEC and EPEC strains. Healthy cattle, sheep, pigs and goats usually transmit these organisms asymptomatically, but it is possible to develop clinical disease. A mixed culture of beta-haemolytic and non-haemolytic *E. coli* strains was also isolated from alpacas. The non-haemolytic *E. coli* produced cytotoxic necrotizing factor (CNF) [74]. Co-infections increase the severity of the disease [5].

A risk factor for *E. coli* infections may be the failure of passive immune transfer with colostrum. Crias lacking maternal antibodies risk dying from colisepticaemia within the first 3–4 days of life. With the partial failure of immune transfer, the risk of colisepticaemia and death is lower, but diarrhoea, pneumonia or chronic arthritis may develop [23]. Colisepticaemia can also occur secondary to other gastrointestinal diseases, such as viral enteritis in older animals [60]. Other risk factors include premature birth, maternal illness or dystocia [22]. Ruminants are a reservoir of STEC. Therefore, keeping NWCs with cattle, diet, direct contact with soil and water contaminated with cattle faeces, and stress may facilitate the transmission of these pathogens [75].

EPEC and EHEC pathotypes usually cause severe, watery, yellowish to brownish and even bloody diarrhoea. The EHEC pathotype is usually isolated from fatal cases with haemorrhagic enteritis [73]. STEC strains mainly cause severe, green, watery diarrhoea in NWCs [75]. SIRS should be considered in crias with symptoms of weakness or sucking abnormalities [22]. Typical signs of colisepticaemia in NWCs are profuse diarrhoea lasting up to 5 days, weight loss, abdominal distention, absence of fever (hypothermia), tachypnoea, tachycardia, pica and debility [22,23]. The development of diarrhoea and septicaemia results in severe water loss, electrolytes and bicarbonate. Cria with *E. coli* diarrhoea develop leukopenia with degenerative left-shift neutrophilia [22].

Clinical signs of meningoencephalitis and brain abscessation may include weakness, dehydration, lethargy, anorexia, neurological symptoms such as inability to stand, opisthotonus, and depression. Cria may fail to defecate and may not be interested in suckling their mother [23]. Meningoencephalitis can develop due to sepsis caused by the failure of passive immunoglobulin transfer. On post-mortem examination, purulent lesions are observed in the brainstem. Lesions may also be present in the brain, lungs and joint fluid [79]. Histopathological examination best confirms purulent meningoencephalitis [23]. In the case Tsur et al. [23] described, a pure culture of *E. coli* was obtained from a purulent brainstem exudate. A/E lesions on histopathological examination are characterised by enterocyte effacement and atrophy with blebbing or scalloping associated with intimately adherent coccobacillary organisms. These lesions are usually located in the small or large intestine [74]. In most cases of *E. coli* infection, clinical signs are insufficient for diagnosis. The isolation and identification of the bacteria are required [23]. However, it has been shown that STEC strains can only be isolated from diarrhoeal faeces on the day of onset of diarrhoea and the following day [79].

In cases of passive immunodeficiency in newborn crias suspected of developing sepsis, administering blood or dam serum, antibiotics and supportive treatment, including intravenous fluid therapy, is recommended [23]. Amoxicillin with clavulanic acid is likely to be the antibiotic of choice [38]. Antibiotic treatment should be based on the antibiogram. However, treatment with broad-spectrum antibiotics such as potassium penicillin (22,000 IU/kg b.w. IV every 6 h) or gentamicin (5 mg/kg b.w. IV 1× daily for 5 days) and intravenous fluid therapy should be administered before the culture results, and antibiogram is available [79]. Dolente et al. [22] suggest treatment with ceftiofur sodium, penicillin, ampicillin, gentamicin, amikacin and metronidazole. In addition, they recommend administering balanced isotonic crystalloid fluids IV or performing a plasma transfusion. Rubio-Langre et al. [80] showed that marbofloxacin is effective against *E. coli* infections in llama crias between 3 and 80 days of age at a dose of 5 mg/kg IV.

In *E. coli*, there is no inherent resistance, so antibiotic resistance is always acquired [81]. Antimicrobial resistance studies of *E. coli* isolated from NWCs in central Germany showed that about 40% of isolates showed the presence of genes mediating resistance to less than three classes of antibiotics. In comparison, about 25% had 10 or more resistance genes. The pathogenic risk and antimicrobial resistance situation for *E. coli* in NWCs in this region appears similar to those in other livestock, companions and wildlife [82]. Another study showed resistance to neomycin in 80% of pathogenic and non-pathogenic *E. coli* strains and 25% to oxytetracycline [83]. In Italy, strains isolated from alpacas showed resistance to ampicillin, florfenicol, oxytetracycline and penicillin [81]. In Peru, *E. coli* isolated from diarrhoeal cases showed resistance to ampicillin, novomycin, tetracycline, penicillin and gentamicin. In contrast, strains isolated from NWCs without diarrhoea showed resistance to gentamicin, tetracycline, ampicillin and penicillin [84].

## 7. Disease Caused by *Pasteurella multocida* and *Manheimia haemolytica*

Pasteurellosis refers to infections caused by bacteria of the genera *Pasteurella* and *Manheimia* that occur in humans and animals. Some species or strains of these bacteria can be primary pathogens causing epidemics of pneumonia and septicaemia with high animal mortality. *Pasteurella multocida* is an agent isolated from NWCs with respiratory disease [24]. Respiratory infections result from interactions between multiple factors, such as pneumonic pathogens, host immune status and environmental conditions [25]. Respiratory diseases, especially infectious ones, cause significant losses in camelid production in the Peruvian Andes. These losses are due to their high morbidity and mortality rates. The most significant losses are recorded among animals up to the first month of life [24].

*P. multocida* is part of numerous animal species’ normal upper respiratory tract microbiota. However, the bacterium is responsible for many clinical manifestations in domestic and wild animals and humans [24]. *Pasteurella* spp. were also isolated from the conjunctival sac of healthy alpacas, llamas and guanacos [85]. *P. multocida* is the primary bacterium responsible for pulmonary infections in alpacas, which, together with *M. haemolytica*, is isolated from hyperacute cases of pneumonia, causing death in young animals during calving and stressful situations. Pneumotropic viruses, such as bovine respiratory syncytial virus (BRSV) and parainfluenza-3 virus (PI3), are combined with *P. multocida* and/or *M. haemolytica* in more than 50% of respiratory infections [25]. Diaz et al. [24] detected the presence of antibodies against *M. haemolytica* and *P. multocida* in Argentine llamas. Argentine NWCs synthesise antibodies that recognise different *M. haemolytica* antigens with high avidity. This suggests that these animals are in contact with *M. haemolytica,* causing only subclinical infections. However, these animals may be a reservoir of these pathogens for other animals [24]. *M. haemolytica* can cause laryngitis with local abscess formation and suffocation due to laryngeal oedema [86].

Virulence factors of *P. multocida* include capsule, lipopolysaccharide, *P. multocida* toxin (PMT), a member of the dermonecrotic toxin family, and iron-regulated and iron-acquisition proteins. These are key in pathogenesis and can vary according to infection and host [24]. Factors favouring the development of the disease include stress caused by environmental factors, such as temperature extremes, viral infections and immunosuppression. These result in promoting bacterial infection and developing pneumonia [24]. Viruses can easily predispose to bacterial colonisation by causing epithelial necrosis and hyperplasia, particularly *P. multocida* and *M. haemolytica* [25]. In young alpacas, *P. multocida* was detected in 52% (24/46) of fatal pneumonia cases. *P. multocida* type A, LPS genotype L6, and *toxA*+ and *tbpA*+ were most frequently isolated [24]. In other studies, *M. haemolytica* was isolated in 53% of cria with pneumonia symptoms and *P. multocida* in 37%. Both bacteria were simultaneously present in 23% of cases [87]. On post-mortem examination, various degrees of pathology may be seen, ranging from congestion, pulmonary oedema, and bilateral suppurative pneumonia to severe and extensive necrotising pneumonia with fibrinous pleuritis. More severe lesions may involve both lungs, mainly the anterior abdominal aspects of the cranial lobes, with haemorrhagic fibrous pleuritis [25].

Histopathological examination of severe necrotising bronchopneumonia and pleuritis shows the presence of massive and extensive coagulative necrosis, with numerous bacterial accumulations and mixed inflammatory responses. In moderate suppurative bronchopneumonia, bronchioles, and the alveolar bedding containing an inflammatory exudate that eventually obliterates the lumen of the bronchi, bronchioles and alveoli are observed [25]. A case of laryngeal abscessation due to *M. haemolytica* in an alpaca was reported in the literature [86]. At necropsy, swelling and redness of the mucosa of the left auricular cartilage were noted. It formed a smooth, oval mass that partially obstructed the rima glottidis. Brown necrotic purulent material was present in the interior. Histologically, there was multifocal, locally extensive to coalescent, necrotic, purulent laryngeal mucositis. Numerous colonies of small bacilliform bacteria surrounded by degenerated neutrophils were present [86]. Studies on resistance to selected antibiotics among *P. multocida* and *M. haemolytica* isolates showed that both pathogens resist penicillin and tetracycline [87].

## 8. Disease Caused by *Corynebacterium pseudotuberculosis*

*Corynebacterium pseudotuberculosis* is a Gram-positive, facultatively intracellular, non-motile and non-spore-forming bacterium [27]. Infection with this bacterium in small ruminants is called caseous lymphadenitis (CLA). Infections can also occur in other animals and humans [88]. *C. pseudotuberculosis* was first isolated from NWCs in 1985 and again in a herd in southern Peru in 1993. In alpacas, it causes abscess formation in lymph nodes and other organs, but unlike in sheep and goats, the lungs are not frequently involved [26]. Abscesses can also occur in the mammary gland, and the bacterium can cause mastitis in alpacas [89]. The disease causes weakness and reduced fibre production in alpacas, but the main economic losses are due to the condemnation of affected carcasses [90]. A strain of Cp267 was isolated from a llama in California, USA, which was responsible for forming a submandibular abscess [27]. The prevalence of abscesses in the alpaca herd was 2.1% (84/3943) [89]. Phospholipase D (PLD) is an exotoxin that is the most immunogenic antigen present in the cell wall of this bacterium. It is also considered one of the most important virulence factors due to limiting bacterial opsonisation, increasing vascular permeability and impairing neutrophil chemotaxis towards the site of infection, resulting in bacterial dissemination [26]. Due to the presence of a lipid envelope and intracellularity, the bacterium can evade the body’s natural defence mechanisms and cause infection in the host [91].

*C. pseudotuberculosis* at the entry site induces an inflammatory process and invades macrophages [26]. The bacterium survives and replicates in macrophages, where it is transported to the surrounding lymph nodes and further spread with their help throughout the body. The transmission route is unclear, but transcutaneously via wounds and abrasions is suspected [88]. The possible contribution of aerogenic transmission and the ingestion of organisms found in milk and colostrum are also suspected [91]. Abscesses can occur in the lymph nodes and the liver. In adult alpacas, abscesses tend to be located in the renal lymph nodes and in young alpacas in the superficial lymph nodes [90]. Anderson et al. [91] described the occurrence of superficial abscesses in the neck area adjacent to the eye and submandibularly. Clinical signs of liver abscesses can include emaciation, anorexia, inability to stand, diarrhoea and caudal ventral and udder oedema [88]. Fever and a significant local inflammatory reaction were observed after experimental intradermal inoculation [90]. Fluid may be present in the abdominal cavity. Blood examination may reveal leukocytosis with neutrophilia, toxic left shift, monocytosis, and non-regenerative anaemia [88]. Leukocytosis with experimental infection persisted for 3 days [90]. Histopathological examination showed severe chronic suppurative hepatitis with necrosis, consistent with liver abscessation. In addition, granulomatous lymphadenitis caused by *C. pseudotuberculosis* was found in numerous lymph nodes [88,90]. Superficial abscesses should be opened, cleaned with sterile saline solution, and left open to drain. A systemic antibiotic may also be used [91]. Rapid drainage of the abscess at the site of inoculation and individual sensitivity to infection may result in the absence of internal abscess development [90]. Isolates from alpacas show the highest susceptibility to ampicillin, penicillin G procaine and erythromycin and these antibiotics are recommended for treatment [91]. The ELISA is an easy and practical test for determining antibody responses in infected animals. However, it is important to note that the severity of the lesions is not correlated with the strength of the immune response in infected NWCs. It is suggested that using *C. pseudotuberculosis* exotoxin as an antigen in this test gives better results than other antigens. After experimental infection, antibodies were detectable from 16 to 128 days after inoculation using an ELISA assay [90]. Other serological tests that can be used for diagnosis are the immunodiffusion and haemolysis inhibition tests. *C. pseudotuberculosis* can be cultured on sheep blood agar plates at 35 °C in an atmosphere with 5% carbon dioxide for 48 h. Colonies form a narrow area of beta-haemolysis [91].

There are commercial toxoid vaccines against CLA used in small ruminants. Braga [26] showed that a vaccine containing high doses of *C. pseudotuberculosis* toxin (265 µg/mL) results in an adequate level of protection in NWCs, as demonstrated by experimental infection of these animals and the absence of abscess production. At low doses of toxin (133 µg/mL), the degree of protection was medium, with abscess formation in regional and internal lymph nodes. The cell-wall vaccinated (cell-wall antigen from *C. pseudotuberculosis*) alpacas showed a lower level of protection against abscess formation but no signs of disease [26].

## 9. Conclusions

Several bacteria can cause disease in camelids. Due to the growing interest in the breeding and rearing llamas and alpacas worldwide, especially in North America and Europe, knowledge of bacterial diseases affecting these animals is necessary to protect their health and public health. Few veterinarians specialise strictly in the management of camelid herds. Their knowledge of these species is often based on information gained independently rather than during their studies. In Europe, there is usually no obligatory course on NWC diseases during veterinary studies; however, in Poland, facultative courses are available in some universities. Therefore, this knowledge should be imparted during study, at least to a basic extent.

### Future Direction

Special care should be taken that bacterial diseases in llamas and alpacas can be potential zoonoses or a source of infection for other domestic animal species. Knowledge of the transmission of infections between different domestic animal species and camelids or between camelids and humans is limited. However, potential transmission between these species must be accepted, given the presence of infections with the same bacterial species found in camelids as in cattle, small ruminants, horses, and humans. Tuberculosis is an example of a disease with zoonotic potential occurring in NWCs. In this case, the reservoirs of the disease for llamas and alpacas are known to be cattle and badgers; however, the route of transmission between camelids is unknown [16,43]. A case of cutaneous tuberculosis in a veterinary surgeon who examined and treated an affected alpaca cria has been reported [39], indicating the possibility of transmission of the infection from NWCs to humans; however, there is a lack of precise information on the possible routes of transmission and other forms of the disease that may develop in humans who have direct contact with a diseased individual. Therefore, this should be further investigated, especially considering that TB has been recognised as an emerging disease in the UK [39], presenting a potential risk to human health. There is limited information on the prevalence of bacterial diseases found in llamas and alpacas, especially in Europe. Available data are often very general, presenting information relating to all New World camelids and genera rather than individual bacterial species. Therefore, in order to improve the control of bacterial diseases in these animals, it would be ideal to collect and compile such data for more European countries, among others. The development of more accurate data on the prevalence of bacterial diseases in llamas and alpacas in European countries and North and South America would allow a comparison of the prevalence of these diseases between the Old and New Worlds. Currently, most data come from South America, with a lesser quantity from North America. Most data gaps exist in Europe. For disease caused by *Clostridium* spp., the data on this was analysed by Twomey et al. [15] in England and Wales. This is the only report on the prevalence of these infections in Europe. It provides the information that 2.5% of samples supplied to the Animal Health and Veterinary Laboratories Agency’s (AHVLA’s) diagnostic laboratory network were samples containing material with clostridia. It was reported that the most common cause of enterotoxaemia was *Clostridium perfringens*. There is a lack of detailed information on other clostridial species and types of *C. perfringens*, as well as a breakdown of data by NWC species. A similar situation exists with *Mycobacterium tuberculosis* complex. Twomey et al. [15] showed only the prevalence of tuberculosis caused by *M. bovis* or *M. microti* is 3.2% in England and Wales, without indicating more detailed information. For the other diseases described, the available information is from South and/or North America, and there is a lack of detailed data from Europe and other parts of the world. A comparison of the prevalence of these diseases between the Old and New Worlds, with the current state of knowledge, would be unreliable, mainly due to data relating to small groups of individuals. The only conclusion that can be drawn is that the diseases described occur in both Europe and the Americas. The study also highlights the need for more research and information on specific bacterial disease entities in SACs, especially regarding treatment and diagnosis. Treatment protocols in llamas and alpacas are often adaptations of protocols used in cattle or horses, taking into account only the lower body weight for dosing. Studies providing information on the antibiotic resistance of bacteria isolated from NWCs, the effectiveness of treatment protocols and the pharmacokinetics and pharmacodynamics of drugs in individual NWC species are highly desirable. For all the diseases described, there is a lack of detailed information on prevention and, in particular, vaccination. An example of a commonly used vaccination in alpacas and llamas is a polyvalent clostridial vaccine. However, there is no vaccine registered for use in these animals in Europe, so products developed for other animals are used, extrapolating vaccination regimen guidelines from these. Publishing more case reports would also be an important step towards a good understanding of alpaca and llama bacterial diseases. Specific rules for the prevention of bacterial diseases in these animals should also be developed. Many of the studies need to be updated, and there needs to be more recent information regarding these diseases, which require researchers to pay more attention to these diseases, epidemiology, and diagnostics in future.

## Figures and Tables

**Table 1 animals-14-00045-t001:** Characteristics of selected bacterial diseases in alpacas and llamas.

Disease	Etiologic Agent	Clinical Signs	Diagnostic Methodology	Treatment	Prevention	References
Clostridial diseases Tetanus Enterotoxaemia type A Enterotoxaemia type C Enterotoxaemia type D	*Clostridium tetani* *Clostridium perfringens* Type A *Clostridium perfringens* Type C *Clostridium perfringens* Type D	No data Neurological signs, colic, intestinal gas tympany, shock, death Elevated body temperature, haemorrhagic diarrhoea, depression, cramps Sudden animal death, neurological symptoms	Presence of clinical signs real-time polymerase chain reaction (PCR) Bacterial culture	Treatment depends on the aetiological agent and the disease entity developed (most often antibiotics therapy)	Vaccinations Adequate colostrum intake	[1] [9,10,11,12,13]
Tuberculosis	*Mycobacterium tuberculosis complex* (*Mycobacterium bovis, M. microti*)	Respiratory distress, enlargement of superficial lymph nodes, discharged skin lesions, death	Post-mortem examination and bacterial culture (definitive diagnosis) Single intradermal comparative tuberculin test (SICTT) Lateral-flow rapid test (RT), multiantigen print immunoassay (MAPIA)	No treatment	No specific guidelines	[14,15,16]
Johne’s disease	*Mycobacterium avium* spp. *paratuberculosis* (MAP)	Protein-losing enteropathy with severe weight loss, asymptomatic infections, non-specific symptoms	PCR Bacterial culture Enzyme-linked immunosorbent assay (ELISA) Agar gel immunodiffusion test (AGID) Complement fixation test (CFT) Post-mortem examination	No specific guidelines	Early detection of MAP carriers and shedders Vaccinations Adequate colostrum intake	[17,18]
Streptococcal infections (Alpaca fever)	*Streptococcus equi* spp. *zooepidemicus*	Septic peritonitis and pleuritis, mastitis, septic orchitis, pneumonia	Cerebrospinal fluid (CSF) testing Bacterial culture Post-mortem examination	Systemic antibiotics and intravenous fluid therapy; Drainage of local lesions;	Metaphylactic treatment of the whole flock	[19,20,21]
*Escherichia coli* infections	*Escherichia coli*	Diarrhoea, septicaemia, neurological signs	Bacterial culture	Administering blood or dam serum; Antibiotics therapy (based on the antibiogram) Supportive treatment (intravenous fluid therapy)	Adequate colostrum intake	[22,23]
Pasteurellosis	*Pasteurella multocida* *Manheimia haemolytica*	Pneumonia, laryngeal abscesses	Bacterial culture	Antibiotic therapy NSAIDs	No specific guidelines	[24,25]
Corynebacterial infections	*Corynebacterium pseudotuberculosis*	Abscesses Mastitis, diarrhoea	ELISA Immunodiffusion Haemolysis inhibition test Bacterial culture	No specific guidelines	Vaccination	[26,27]

## Data Availability

No new data were created or analyzed in this study. Data sharing is not applicable to this article.

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
