# Peer review of "A Literature Review of Selected Bacterial Diseases in Alpacas and Llamas—Epidemiology, Clinical Signs and Diagnostics"

_animals, 2023, doi:10.3390/ani14010045_

Round 1

Reviewer 1 Report

Comments and Suggestions for Authors

Dear authors,

The manuscript title is “Selected infectious diseases of New-World Camelids (NWC) – a review of epidemiology, clinical picture and diagnostics” and it aims to review some bacterial diseases found in these animals.

The topic falls within the aims and scope of the journal and if it is such an emergent issue, the review should be of interest. However, the total absence of the methodology used to perform this review is a major concern.

Some particular suggestions/comments will be done here:

- Lines 2/3 – Please do not use acronyms in the title; I also would suggest to write “signs” instead of “picture”; infectious is much more than bacterial diseases, so if the manuscript is on bacterial, maybe you should state it in the title – “bacterial” instead of “infectious”

- Line 25 – again the “picture”

- line 26 – I suggest you not to repeat in keywords words that are already in the title; you may write, for example, the name of some camelids

- Line 34 – no italics Camelidae (you only need them in species and genus)

- Lines 34/35 – I think Lamini and Camelini are not in italics again

- Line 56 – please write in full ADHD as it is the first place

- Line 62/63 – how did you define which one are the most important bacterial diseases? Please delete infectious as all bacterial diseases are infectious.

- Line 66 – what do you mean with “selected”? You must state how did you do the “selection”. Why did you select these and not other bacterial diseases? You need also to describe the methodology used to perform this review, that is an essential point.

Table 1:

-      Please, separate treatment and prevention in 2 columns.

-      Write only “Disease” in the head

-      Etiologic agent instead of factor

-      Diagnostic methodology

-      Treatment

-      Signs instead of symptoms

-      First time PCR, ELISA and others, write full please

-      spp is not in italics

-      do you mean different things with abscessation and abscesses? If not please be coherent

-      What is the interest of this table if you are going to develop after these diseases?

-      Line 75 – 77 – Clostridium spp has very different species with very different health consequences and some are part of the animal microflora, what is the interest of this information if you do not identify which species were identified and not even in what kind of samples?

-      Line 114/164/326 and other places, please review all – CNS/UK/US write in full, first time

-      Line 148 – infections instead of infestations

-      Line 275/649/review all the manuscript – necropsy instead of autopsy

-      Line 418 – brain?

-      Line 430 – Streptococcus

-      Lines 429 – 451 – you used always the same reference, this may be consider plagiarism and it is not for sure a review

-      Lines 485/538/615 – first you have to write full and then (MIC/CNF/BRSV)

-      Conclusion: did you find any difference between the occurrence of the different diseases reviewed in the new and the old world?

Author Response

Reviewer 1

Dear authors,

 The manuscript title is “Selected infectious diseases of New-World Camelids (NWC) – a review of epidemiology, clinical picture and diagnostics” and it aims to review some bacterial diseases found in these animals.

The topic falls within the aims and scope of the journal and if it is such an emergent issue, the review should be of interest. However, the total absence of the methodology used to perform this review is a major concern.

Some particular suggestions/comments will be done here.

AU: Thank you for carefully reading our manuscript and for your precious comments and suggestions. With these improvements, we hope that the current version of the manuscript can meet the Journal’s standards for publication.

Specific comments:

- Lines 2/3 – Please do not use acronyms in the title; I also would suggest to write “signs” instead of “picture”; infectious is much more than bacterial diseases, so if the manuscript is on bacterial, maybe you should state it in the title – “bacterial” instead of “infectious”

AU: Thank you for a valuable comment. We have changed the suggested words in lines 2/3. The acronym from the title has been removed.

- Line 25 – again the “picture”

AU: Thank you for a valuable comment. We have changed the word “picture” to “signs” in line 26.

- Line 26 – I suggest you not to repeat in keywords words that are already in the title; you may write, for example, the name of some camelid

AU: Thank you for a valuable comment. We have added new keywords as suggested. Lines 27/28.

- Line 34 – no italics Camelidae (you only need them in species and genus)

AU: Thank you for a valuable comment. We have introduced a change in the line 36.

- Lines 34/35 – I think Lamini and Camelini are not in italics again

AU: Thank you for a valuable comment. Done as requested. Lines 36/37.

- Line 56 – please write in full ADHD as it is the first place

AU: Thank you for your valuable comment. Done as requested. Lines 58/59.

- Line 62/63 – how did you define which one are the most important bacterial diseases? Please delete infectious as all bacterial diseases are infectious.

AU: Thank you for your valuable comment. We have changed words “the most important” to “selected”. In choosing bacterial diseases, we were guided by the quantity of literature available on them. We have deleted word “infectious”. Line 77 and 79. We have also added a methodology for the selection of articles used in the publication and criteria for the selection of diseases. Lines 65-76.

- Line 66 – what do you mean with “selected”? You must state how did you do the “selection”. Why did you select these and not other bacterial diseases? You need also to describe the methodology used to perform this review, that is an essential point.

AU: Thank you for your valuable comment. We have added the methodology used to perform this review. Line 65-72. We have provided a methodology for selecting the diseases described in our review. Lines 72-76.

- Table 1:

Please, separate treatment and prevention in 2 columns.

AU: Thank you for your valuable comment. Done as requested. Table 1.

Write only “Disease” in the head

AU: Thank you for your valuable comment. Done as requested. Table 1.

Etiologic agent instead of factor

AU: Thank you for your valuable comment. Done as requested. Table 1.

Diagnostic methodology

AU: Thank you for your valuable comment. Done as requested. Table 1.

Treatment

AU: Thank you for your valuable comment. Done as requested. Table 1.

Signs instead of symptoms

AU: Thank you for your valuable comment. Done as requested. Table 1.

First time PCR, ELISA and others, write full please

AU: Thank you for your valuable comment. Done as requested. Table 1.

spp is not in italics

AU: Thank you for your valuable comment. Done as requested. Table 1.

do you mean different things with abscessation and abscesses? If not please be coherent

AU: Thank you for your valuable comment. We changed “abscessation” to “abscesses”. Table 1.

What is the interest of this table if you are going to develop after these diseases?

AU: Thank you for your valuable comments. Done as requested. Table 1. The table here is intended to give the reader an idea of exactly which diseases are included in the review. In addition, the information in the table allows the reader to quickly find key information about these diseases. Later in the review, the reader can expand his or her knowledge with detailed information available in the scientific literature.

- Line 75 – 77 – Clostridium spp has very different species with very different health consequences and some are part of the animal microflora, what is the interest of this information if you do not identify which species were identified and not even in what kind of samples?

AU: Thank you for your valuable comment. We agree with the suggestion. In the literature, data on bacterial diseases in camelids are very sparse. We have used all available information. These data are from an article about laboratory submissions from NWCs in England and Wales from 2000 to 2011. This review only reported the number of cases of clostridiosis identified, of which enterotoxaemia was the most commonly diagnosed disease. Individual species were not identified. We have completed the missing information in the text. Lines 104-106.

- Line 114/164/326 and other places, please review all – CNS/UK/US write in full, first time

AU: Thank you for your valuable comment. For each abbreviation that appears for the first time in the text, we have given its full meaning. Lines 143, 193, 354, 357, 407.

- Line 148 – infections instead of infestations

AU: Thank you for your valuable suggestion. Done as requested. Line 177.

- Line 275/649/review all the manuscript – necropsy instead of autopsy

AU: Thank you for your valuable comment. Done as requested. Lines 305/686.

- Line 418 – brain?

AU: We apologise for any confusion. Of course, it should be the word 'brain'. Line 450.

- Line 430 – Streptococcus

AU: Thank you for your valuable comment. Done as requested. Line 461.

- Lines 429 – 451 – you used always the same reference, this may be consider plagiarism and it is not for sure a review

AU: Thank you for your valuable suggestion. In this section of the text, we have not included all the references used. We have added the missing citations. Lines 461-485 We have also completed the missing references. Lines 945-956.

- Lines 485/538/615 – first you have to write full and then (MIC/CNF/BRSV)

AU: We apologise for any confusion. Done as requested. Lines 519/520, 524/525, 573/574, 651/652.

- Conclusion: did you find any difference between the occurrence of the different diseases reviewed in the new and the old world?

AU: Thank you for your valuable comment. We have added the relevant fragment in lines 767-781.

Reviewer 2 Report

Comments and Suggestions for Authors

It is a very useful manuscript. Ensure you observe the Journal format.

Comments on the Quality of English Language

The English is easy to read and comprehend.

Author Response

Reviewer 2

Review Report for article: Selected infectious diseases of New-World Camelids (NWC) – a review of epidemiology, clinical picture and diagnostics

A brief summary

The aim of this article was to review and gather the latest knowledge on selected bacterial diseases of New-World Camelids, as well as to identify information deficiencies regarding epidemiology, clinical picture and diagnostics for breeders and veterinarians. Because the breeding of South American Camelids (SACs), including, llamas, alpacas, guanacos and vicuñas is becoming increasingly popular in regions that are not their natural habitat, especially in Europe. These camelids are considered to be relatively disease-resistant. However, due to their growing popularity, special attention should be given to infections in these animal species, especially infectious diseases, many of which are zoonotic. There is limited information on many diseases occurring in llamas and alpacas, making veterinarians rely on data developed in other animal species, such as cattle, sheep and horses. This review on bacterial diseases occurring in SACs, with particular attention to epidemiology, clinical picture and diagnostics will be very useful.

General concept comments

Article: highlighting areas of weakness, the testability of the hypothesis, methodological inaccuracies, missingcontrols, etc.

The article presents an extensive review on epidemiology, clinical picture and diagnostics of bacterial infections of south American Camelids that are increasing becoming popular in Europe, given their disease-resistance traits.

Review: commenting on the completeness of the review topic covered, the relevance of the review topic, the gap in knowledge identified, the appropriateness of references, etc. These comments are focused on the scientific content of the manuscript and should be specific enough for the authors to be able to respond.

The article presents exhaustive review data on the subject matter.

Specific comments referring to line numbers, tables or figures that point out inaccuracies within the text or sentences that are unclear. These comments should also focus on the scientific content and not on spelling, formatting or English language problems, as these can be addressed at a later stage by our internal staff.

The article is a significant contribution to Journal animal. It should be accepted for publication. Given its strength of scientific data.

AU: Thank you for carefully reading our manuscript and for your valuable opinion. We believe that our review will contribute to raising scientists' awareness of the need to improve knowledge of bacterial diseases of New World camelids.

Reviewer 3 Report

Comments and Suggestions for Authors

Dear sir

In this review not all bacterial disease affecting camelids have been covered and there was no orderly writing of different category of camelids. The literature review are incomplete and need through reviewing of all the published articles and need extensive editing of the article.

Author Response

Reviewer 3

 A brief summary

This article planned to provide a bacterial disease of the new old camelids and though author attempted for compilation but it is not up-to-date.

AU: Thank you for carefully reading our manuscript and for your valuable comments and suggestions. The aim of our article was to provide an overview on selected bacterial diseases of New World camelids only. Old World camelids were deliberately not included in this review. We focused on NWCs because of their increasing popularity especially in the United States and Europe and the zoonotic potential of some of them. In addition, the literature on bacterial diseases in OWCs is extensive enough to constitute a separate review. The methodology used in the search for the most up-to-date data on these diseases is added in the lines 65-76.

General comments Not all bacterial disease affecting camelids have been covered

No orderly writing of different category of camelids

The literature review are incomplete and need through reviewing of all the published articles and need extensive editing

AU: Thank you for your precious suggestions. Our aim was not to describe all bacterial diseases of New World camelids. Bacterial diseases of camelids are plentiful and the information about them available in the literature could be material for a chapter in a book. We wanted to collect and organise knowledge on selected diseases about which there is the most information.

A description of each disease with a chapter on all New World camelid species is currently not possible, due to the very scarce data on vicuña and guanaco. There is only a single piece of information on them, not every disease has been described in them, and the vast majority of information is from South America, particularly Peru. 

In preparing the manuscript, we used the latest knowledge using search engines valued by researchers worldwide. A description of the methodology has been added in lines 65-76.

More tables may be compiled and all the Camelids dromedary camels, Bactrian camels, wild Bactrian camels, llamas, alpacas, vicuñas, and guanacos may be covered

AU: Thank you for your valuable suggestions. As we mentioned above, our review is only concerned with New World camelids A description of each disease with a distribution across all New World camelid species is currently not possible, due to the very sparse data on vicuña and guanaco. Thus, creating a table comparing individual species of all camelids is impossible, or would require the creation of a meta-analysis. Our idea was to create an overview.

More diseases affecting camelids viz., bacterial, viruses and parasitic may be compiled

AU: Thank you for your comment. Including diseases of other than bacterial aetiology is not the purpose of our review. A review of knowledge on viral and parasitic diseases is a good idea for separate articles. In addition, the inclusion of all bacterial diseases of camelids in the review would create a manuscript that is too long for the standard reader and would reduce its attractiveness.

Etiology; Camelids family affecting; Causative organizing; Specific sign; Differential diagnosis; Family specific variation in disease pattern; Preventive measures; Treatment options may be arranged for each diseases

AU: Thank you for your suggestion. The proposed disease description scheme was applied in our review. The lack of information on bacterial diseases of NWCs prevents an accurate description of each disease. Identifying information gaps is one of the objectives of our review.

The review coverage is not up-to-date and no detailed explanation of the gap in knowledge identified has been given. Some references are not appropriate

AU: Thank you for your valuable comment. The aim of our review was to gather all available information on selected bacterial diseases of New World camelids. In collecting the information, we used the most up-to-date scientific knowledge. We used search engines such as Google Scholar and PubMed. In the lines 762-792 a gap in knowledge about the diseases described in the review has been clarified.

Diseases specific to this region may also be clearly highlighted

AU: Thank you for your valuable suggestion. A lot of information on the incidence of bacterial disease in NWCs refers to studies conducted in North America and South America. Few data are published from Europe. This is one of the knowledge gaps that should be filled in order to better protect humans and animals. Therefore, highlighting diseases specific to Europe is difficult. However, in selecting the diseases described in the review, we have tried to include diseases that pose a threat in all New World camelid locations.

Disease outbreaks, mortality and morbidity percentages of all the diseases for all camelids family may be given

AU: Thank you for your valuable suggestion. Available information on recorded cases of the diseases in the literature, as well as available morbidity and mortality data, is provided when describing each disease, for example the lines 195-203, 379-386, 553-562.

The diseases like Anthrax. Tuberculosis. Salmonella. Pasteurellosis. Paratuberculosis (Johne's Disease).. Clostridial Diseases. Brucella and Streptococcus spp. , Klebsiella, E. coli, Enterococci, Bacillus, Corynebacterium spp., Staphylococcus aureus, and Streptococcus pyogenes Cutaneous Fungal Infection etc may be covered for each camelids groups.

AU: Thank you for your valuable suggestion. As we mentioned earlier, expanding the review to include additional diseases would lose its appeal to the reader due to its length.

Table 1. Is all the diseases listed and the clinical signs specific to came may be given and the treatment column may be deleted as it is not informative

AU: Thank you for your valuable suggestion. The purpose of creating Table 1 was to collect key data on each of the diseases described, including etiology, clinical signs, diagnostic methodology, treatment/prevention. We decided to separate the treatment/prevention column. Table 1. It seems to us that leaving the treatment column is a good representation of the lack of precise guidelines for the treatment of bacterial diseases in NWCs.

Line 76. Clostridia were detected in approximately 2,5% of the samples 76 (44/1765).

AU: Thank you for your comment. The studies cited do not state which exact NWCs species were involved. In the lines 102/103 we have indicated that the data refer to NWCs in general.

Line 131. The results of this study indicate – The results of this study indicated.

AU: Thank you for your comment. Done as requested. Lines 160/161.

Line 69-154. Most of the information provided are general and may be modified with specific to camelids.

AU: Thank you for your suggestion. On lines 96-114 the general characteristics of the bacteria of the genus Clostridium spp. are described. This general introduction occurs with all the diseases described in the review. In the lines 115-183 we have described information on Clostridium perfringens types A, C and D and other clostridial infections, possible diagnostic methods and the vaccination regimen for New World camelids. The data used to create this paragraph are taken from specialised articles on NWCs. 

Line 193. 0,1 ml bovine – 0.1 ml bovine.

AU: Thank you for your comment. Done as requested. Line 223.

Round 2

Reviewer 1 Report

Comments and Suggestions for Authors

Dear authors,

Congratulations on the improvements.

I have doubts about the criterion used to select the diseases and I still believe Table 1 is totally unnecessary, but I leave these decisions to the Editor.

Best regards

Author Response

 Thank you very much for your positive feedback on the improvements we made 

Reviewer 3 Report

Comments and Suggestions for Authors

the article has been modified as suggested and may be considered fro publication

Author Response

Thank you for this opinion